# Development of a Dual-Vector System Utilizing MicroRNA Mimics of the *Autographa californica* miR-1 for an Inducible Knockdown in Insect Cells

**DOI:** 10.3390/ijms20030533

**Published:** 2019-01-27

**Authors:** Krisztina Koczka, Wolfgang Ernst, Dieter Palmberger, Miriam Klausberger, Lisa Nika, Reingard Grabherr

**Affiliations:** 1Austrian Centre of Industrial Biotechnology - acib, A-1190 Vienna, Austria; krisztina.koczka@boku.ac.at (K.K.); wolfgang.ernst@boku.ac.at (W.E.); dieter.palmberger@boku.ac.at (D.P.); miriam.klausberger@boku.ac.at (M.K.); 2Department of Biotechnology, University of Natural Resources and Life Sciences, A-1190 Vienna, Austria; lisa.nika@boku.ac.at

**Keywords:** artificial miRNA, baculovirus-insect cell expression system, RNA interference

## Abstract

The baculovirus-insect cell expression system is a popular tool for the manufacturing of various attractive recombinant products. Over the years, several attempts have been made to engineer and further improve this production platform by targeting host or baculoviral genes by RNA interference. In this study, an inducible knockdown system was established in insect (*Sf*9) cells by combining an artificial microRNA precursor mimic of baculoviral origin and the bacteriophage T7 transcription machinery. Four structurally different artificial precursor constructs were created and tested in a screening assay. The most efficient artificial microRNA construct resulted in a 69% reduction in the fluorescence intensity of the target enhanced yellow fluorescent protein (eYFP). Next, recombinant baculoviruses were created carrying either the selected artificial precursor mimic under the transcriptional control of the T7 promoter or solely the T7 RNA polymerase under a baculoviral promoter. Upon co-infecting *Sf*9 cells with these two viruses, the fluorescence intensity of eYFP was suppressed by ~30–40% on the protein level. The reduction in the target mRNA level was demonstrated with real-time quantitative PCR. The presented inducible knockdown system may serve as an important and valuable tool for basic baculovirus-insect cell research and for the improvement of production processes using this platform.

## 1. Introduction

The baculovirus-insect cell expression system (BEVS) is widely used for the manufacturing of various recombinant products, e.g., virus-like particles to be applied as vaccines [1] or display scaffolds [2], viral vectors for gene therapy [3], gene delivery vectors [4], and various other complex proteins. The popularity of this platform lies in its advantageous properties that allow for high product titers in combination with lower production costs as compared to production in mammalian cells [5] as well as proper post-translational modifications such as glycosylation [6]. Additionally, several attempts have been made over the years to make this expression system even more appealing for industrial use. Besides the establishment of efficient expression vectors along with easy transfection and cloning procedures, RNA interference (RNAi) has been exploited to specifically downregulate selected genes during the production process.

RNAi is a conserved biological process using short RNA molecules that include small-interfering RNAs (siRNAs) and microRNAs (miRNAs) for the regulation of gene expression [7]. siRNAs are ~21–25 nucleotide (nt) long non-coding double-stranded RNA (dsRNA) molecules that are produced in response to foreign nucleic acid originating from exogenous invaders (such as viruses or transposons) to sequence-specifically degrade perfectly complementary target messenger RNAs (mRNAs) [8]. In contrast, miRNAs are ~19–24 nt long non-coding RNA pieces originating from purposefully expressed stem-loop precursors with an incomplete double-stranded character, called primary miRNA (pri-miRNA) transcripts. Unlike siRNAs, miRNAs show imperfect complementarity to their target mRNA sequences, and they mediate translational repression and transcript degradation [9].

In general, several studies have been carried out in insect cells focusing on the exploitation of the RNAi pathway to create a tool for gene function studies and to improve the production system by specific regulation of host or baculoviral gene expression. Huang et al. [10] established a baculovirus-based system in *Spodoptera frugiperda Sf*9 cells to achieve the synthesis of long hairpin dsRNAs in vitro that might serve as a reverse genetics tool. Although long dsRNAs are easy to design and they abolish the expression of a target gene effectively, the processing of a long hairpin may result in multiple siRNA duplexes targeting various genes, which carries the risk of severe off-target effects [11]. Another, possibly more target-specific approach utilizes short hairpin RNAs (shRNAs). Studies [12] report the successful downregulation of *N*-acetylglucosaminidase to generate more mammalian-like glycosylation in *Trichoplusia ni* cells using shRNAs delivered by DNA vectors. Others [13] aimed at an improved production platform by targeting *S. frugiperda* caspase-1 using a baculoviral shRNA vector. However, in order to obtain well-defined transcript termination, the expression of shRNAs in vivo necessitates the use of strong Pol III promoters, which not only limits the application of the system but may also cause cytotoxicity in the host cells [14].

Alternatively, synthetic siRNAs that target a chosen gene can be embedded within endogenous pri-miRNA transcripts that serve as backbones. The expression of such mimics can be driven by Pol II promoters to overcome the cytotoxicity linked to Pol III promoters [14]. The pri-miRNA carrying the synthetic siRNA—or the so-called artificial microRNA (amiRNA)—mimics the natural transcript and is recognized and processed by the host cell´s microRNA biogenesis pathway [15]. Briefly, concurrently with transcription by RNA polymerase II, the natural pri-miRNA transcript (or a mimic) is cleaved by the microprocessor complex comprising one molecule of Drosha and two molecules of its cofactor Pasha [16]. The resulting ~70 nt long precursor hairpin structure, known as the pre-miRNA, is thereafter exported to the cytoplasm by the transporter protein Exportin-5 together with Ran-GTP. Here, another processing complex consisting of Dicer-2 and its cofactors the dsRNA-binding proteins Loquacious PD isoform (Loqs-PD) and R2D2 cleaves the pre-miRNA to short double-stranded RNA fragments [17]. Next, through the binding of Argonaute 2 (Ago2) protein with the RNA duplex, the precursor RNAi-induced silencing complex (pre-RISC) is formed [18]. The mature RISC contains only the strand with the less stable base pair (bp) at the 5´ end, called the guide strand, whereas the other strand of the RNA duplex (passenger strand) is cleaved. When a pri-miRNA transcript mimic is processed, the customized amiRNA sequence is loaded into the mature complex [15]. Finally, this activated multi-enzyme complex cleaves complementary mRNA sequences [7]. Haley et al. [11] established such a system in *Drosophila melanogaster* S2 cells and transgenic flies. Moreover, Zhang et al. [19] reported that the use of *Bombyx mori* nucleopolyhedrovirus pri-miRNA mimics to effectively inhibit viral replication in silkworm.

The pri-miRNA transcript of *Autographa californica* nucleopolyhedrovirus miR-1 (*Ac*MNPV-pri-miR-1) is the first and, to our knowledge, so far only miRNA discovered in the genome of *Ac*MNPV [20]. To reveal and assess its silencing capabilities as a pri-miRNA mimic backbone, the original siRNA duplex within the stem-loop structure was replaced by a synthetic sequence targeting eYFP. Additionally, four different structural versions of the natural transcript were created by applying small changes in the stem sequences and the flanking regions according to general design rules [14]. To evaluate the silencing potency of these customized constructs, a plasmid-based screening was carried out. According to this data, the most effective stem-loop structure was selected for insertion into the genome of *Ac*MNPV. For the transcriptional control of the most effective silencer mimic, a T7-based expression system was developed. The bacteriophage T7 transcription machinery is an attractive system due to its strict promoter selectivity and high catalytic activity [21]. Several successful attempts have been made in past years with the aim of utilizing this 100 kDa prokaryotic enzyme for protein expression in eukaryotic cells, including mammalian [22], insect [23,24], and plant [25] cell lines.

The experimental setup presented here relies on two viral vectors. One recombinant baculovirus was designed to harbor an amiRNA construct linked to the bacteriophage T7 promoter, while the corresponding T7 RNA polymerase (T7 RNAP) with an additional nuclear localization signal was expressed by a second viral vector. The selective transcriptional activity, together with the fact that the functional expression of the amiRNA construct depends on the presence of the T7 RNAP that is encoded on a separate virus constitute an inducible system. Even when essential genes are targeted for downregulation, the two viral expression vectors can be produced efficiently whenever they are separate from each other. We could show that the T7 RNAP is functional in *Sf*9 cells as it activates the transcription of genes that are under control of the T7 promoter. We further demonstrated that amiRNAs are functional after transcription by the T7 RNAP as they successfully downregulated the reporter gene eYFP. Our T7-based inducible expression system may serve as a valuable tool for gene regulation during a production process, e.g., by altering the glycosylation pattern or downregulating essential genes such as proteases or proteins involved in baculovirus assembly.

## 2. Results and Discussion

### 2.1. amiRNA Construct Design

Here we describe a novel, inducible knockdown system established in *Sf*9 cells. Our approach is based on combining the transcription machinery of the prokaryotic bacteriophage T7 [23] with RNAi-based amiRNA constructs derived from the *Ac*MNPV-pri-miR-1 transcript [20]. One common way to establish a targeted RNAi system is to exploit the endogenous miRNA biogenesis pathway by embedding a small, artificial RNA molecule into the hairpin structure of a natural pri-miRNA transcript. This method is considered more advantageous over the traditional siRNA-based approaches, as it confers higher gene silencing efficiency in combination with reduced off-target effects [26]. Since baculoviruses take control of the host cell synthesis machinery as part of the infectious cycle and downregulate almost all cellular genes [27], it was of importance to select a viral miRNA precursor, which is highly abundant during the infection cycle. Thus, the 58 nt long precursor hairpin of the baculovirus-encoded *Ac*MNPV-miR-1 (*Ac*MNPV-pre-miR-1) was selected. The 20 nt long processed miR-1 targets the viral gene *ac94* (ODV-E25) [28]. In our experimental setup, the eYFP was chosen as a model target for the amiRNA. Therefore, each hairpin harbored a highly effective, previously described customized siRNA sequence targeting enhanced green fluorescent protein (eGFP) [29]. Notwithstanding that a slightly different eGFP variant was applied in this study, the synthetic duplex embedded in the amiRNA constructs was still applicable to estimate the silencing capacity of the proposed system, as the eYFP sequence used in these experiments also contained the target nucleotides of the published synthetic siRNA.

Overall, four artificial versions of the *Ac*MNPV-pri-miR-1 transcript were generated: amiR-1A, amiR-1B, amiR-1C, and amiR-1D (amiR-1A-D). Sequences of the amiRNA constructs are listed in Table 1, and the precursor hairpin structures (without the flanking regions) are presented in Figure 1. The 58 nt long amiR-1A hairpin construct mimics the natural *Ac*MNPV-miR-1 in all aspects, as they share the same internal structure and the nucleotide composition of the pre-miRNA backbone stem and loop. Moreover, amiR-1A contained 31 nt long flanking regions of the natural *Ac*MNPV-pri-miR-1 transcript upstream and downstream of the hairpin without internal restriction sites, since the construct was synthesized as a single piece of DNA fragment. The processing of the artificial construct by the miRNA biogenesis pathway should result in a 20 nt long guide strand. This small RNA contains a base pair mismatch to the target eYFP sequence at the terminal 3’ nucleotide (T→C). Thereby, the 3’-end becomes more stably paired to the passenger strand (in comparison to the 5’ end starting with A) and facilitates its incorporation into the RISC. Additionally, a mismatch at the 3’ end of the guide strand was shown to be advantageous, as it induces more potent silencing in mammalian cells [30].

Based on the effectiveness of similar setups [11,31], the internal structure of the stem-loop constructs amiR-1B-D was substantially different from that of amiR-1A (Figure 1), as the stem region of these hairpins showed perfect complementarity, thereby lacking the bulges and mismatches that can be found in the pre-miR-1. Furthermore, the longer flanking regions of 120 nt upstream and downstream of the stems necessitated a different, but commonly used cloning procedure (see Section 3.2), where the insertion of the hairpins, as annealed oligonucleotides (oligos), was facilitated by restriction sites between the flanking regions, resulting in short scar sequences between the hairpin’s ends and the flanking sequences. This, however, did not seem to interfere with the processing of the amiRNA as shown for the amiR-1C construct (see Appendix A section “Detection of Mature amiRNAs”). Apart from these differences, the 56 nt long amiR-1B construct shares the exact same embedded 20 nt guide strand with the amiR-1A hairpin structure, including the above-mentioned mismatch at Position 20. The amiR-1C and amiR-1D constructs were modified based on previous findings regarding the two possible mechanisms of cleavage site recognition of the microprocessor complex (consisting of Drosha and its cofactor Pasha) in the early steps of the miRNA biogenesis pathway. According to this data [32,33,34], the dominant process is that, with Pasha serving as a molecular ruler, the enzyme complex cuts 11 nt distant from the basal junction along the stem of the pri-miRNA hairpin structure. In contrast, the less governing cleavage mechanism initiates the cleavage 22 nt distant from the apical junction. Theoretically, in order to achieve precise cleavage, pri-miRNA transcript mimics should be designed in such a way that both mechanisms lead to the same cleavage sites. However, neither the natural hairpin structure of *Ac*MNPV-pri-miR-1 nor the mimic constructs amiR-1A or amiR-1B fulfills the 11 nt or the 22 nt distance criterion mentioned above. The amiR-1C and amiR-1D stem-loop constructs were nevertheless designed with regard to these distance constraints. The guide strand embedded in the 60 nt long amiR-1C hairpin was identical to the original 21 nt long siRNA published previously [29]. It contained no mismatches to the target eYFP (as opposed to amiR-1A-B), and an extra C-G bp was inserted into the stem region of the construct between the artificial RNA duplex and the loop. These two modifications increase the distance between the basal and apical junctions to 22 nt, making the structure accessible to the secondary microprocessor cleavage mechanism. The 72 nt long amiR-1D construct includes the complete hairpin of the amiR-1C stem-loop structure with an additional 6 nt stem extension to increase the distance between the basal junction and the required cut site to 11 nt. Thus, regardless of the preferred cleavage process of the microprocessor complex, the processing of the amiR-1D construct should lead to the same precursor hairpin product. For the precise assessment of the four constructs (amiR-1A-D), controls with matching structures and scrambled embedded siRNA sequences were created: amiR-1As, amiR-1Bs, amiR-1Cs, and amiR-1Ds (amiR-1As-Ds).

### 2.2. Plasmid-Based Evaluation of the amiRNA Constructs

A plasmid-based screening assay was carried out, to evaluate and select the best amiRNA construct. *Sf*9 cells seeded into 6-well plates were transfected with 200 ng of the reporter plasmid pACEBac1ie1-eYFP in combination with 2 µg of one of the eight following plasmids: pACEBac1ie1amiR-1A-D or pACEBac1ie1amiR-1As-Ds (control). Forty-eight hours post-transfection (h p.t.), the silencing efficiency was evaluated by microscopy and flow cytometry. Figure 2 shows the matching phase contrast and fluorescence images of the co-transfections. A merely visual estimation of the images suggests that the strongest silencing on the protein level was achieved (in comparison to the structure-specific scrambled controls), when either the amiR-1B or amiR-1C construct was applied, whereas in the case of amiR-1A or amiR-1D hardly any difference could be observed between the control and sample fluorescence images. These results were further supported by the data obtained by flow cytometry measurements (Figure 2). As a basis of comparison, the sum of intensity (overall fluorescence intensity) values was calculated by multiplying the fluorescent cell count with the arithmetic mean of the fluorescence intensity. According to this, the amiR-1B and amiR-1C hairpin constructs were the most efficient mimics in silencing the target gene eYFP, with 59% and 69% reductions in the overall fluorescence, respectively. The increased distance between the hairpin’s apical and basal junctions in amiR-1C might have increased the processing efficiency by the microprocessor complex, leading to a greater decrease in eYFP fluorescence intensity as compared to amir-1B. In contrast, the lowest efficiency of 14% reduction in eYFP intensity was achieved with amiR-1D, containing the most modifications as compared to the original *Ac*MNPV-miR-1 precursor structure. This suggests that certain changes or too many modifications are not tolerated by the endogenous miRNA processing pathway. Surprisingly, the amiR-1A construct achieved only a 20% reduction in eYFP intensity, despite a high resemblance to the original *Ac*MNPV-miR-1 precursor hairpin. A reason behind this could be the rather short flanking regions included in this construct. As the amiR-1A mimic contains only 31 nt single stranded flanking RNA upstream and downstream of the hairpin structure, it is possible that this has an impact on the secondary structure of the construct. Based on these results, the amiR-1C hairpin construct was considered most effective for the downregulation of the target eYFP with the embedded amiRNA and was selected for further studies. However, it must be noted that the embedded amiRNA might affect its processing in the hairpin; thus, one of the other constructs may prove to be more effective in combination with other amiRNA sequences.

### 2.3. The Inducible Knockdown System

The amiR-1C hairpin structure was selected based on the preliminary screening experiments as the most efficient gene silencer. The baculoviruses *Ac*-T7amiR-1C_ ie1eYFP and *Ac*-T7amiR-1Cs_ ie1eYFP harboring the eYFP under the control of the ie1 promoter, together with either the amiR-1C or the amiR-1Cs pri-miRNA under the control of T7 promoter, respectively. The inducibility of the system lies in the fact that, in the absence of the T7 RNAP, the hairpin structures are not transcribed, as the T7 promoter is inactive without its corresponding T7 polymerase. Therefore, the *Ac*-ie1T7RNAP virus served as the inducer of the silencing effect. Before the setup of the inducible system, the expression and the functionality of the *Ac*-ie1T7RNAP virus was confirmed with immunoblotting and an in vitro transcription assay (see Figure A1). The system’s mechanism of action is illustrated in Figure 3. Briefly, for the transcription of a pri-miRNA transcript mimic and the subsequent production of a mature amiRNA, the simultaneous presence of either the *Ac*-T7amiR-1C_ ie1eYFP or the *Ac*-T7amiR-1Cs_ ie1eYFP (control) baculovirus (Virus A) and the *Ac*-ie1T7RNAP virus (Virus B) is necessary. Upon co-infection of *Sf*9 cells with the two recombinant baculoviruses, the T7 RNAP expressed from Virus B transcribes the pri-miRNA transcript mimic encoded by Virus A. The entry of this transcript into the host’s endogenous miRNA processing pathway is followed by the digestion with the Drosha and Dicer-2 nucleases, resulting in an amiRNA duplex containing either an eYFP targeting (amiR-1C) or a scrambled control (amiR-1Cs) guide amiRNA strand. The mature RISC containing only the guide amiRNA strand initiates the degradation of perfectly complementary mRNA sequences and thereby impedes translation and product formation of eYFP (provided that the *Ac*-T7amiR-1C_ ie1eYFP virus was used as Virus A).

*Sf*9 cells were co-infected at various multiplicity of infection (MOI) combinations with the *Ac*-ie1T7RNAP virus and either the *Ac*-T7amiR-1C_ ie1eYFP or the *Ac*-T7amiR-1Cs_ ie1eYFP (control) baculovirus. Samples were collected 48 h post-infection (h p.i.) to evaluate the silencing efficiency of the selected artificial hairpin structure (amiR-1C) in comparison to its scrambled sequence control (amiR-1Cs) on the protein level. Figure 4 shows the flow cytometry results of co-infections, where the MOI of the *Ac*-T7amiR-1C_ ie1eYFP and *Ac*-T7amiR-1Cs_ ie1eYFP (control) viruses was either 1 or 5, whereas the *Ac*-ie1T7RNAP was added to the cultures at either MOI 5 or MOI 10. The 5-fold excess of the *Ac*-ie1T7RNAP virus resulted in a 37% decrease in the overall eYFP fluorescence intensity, whereas upon applying a 10-fold excess, a 31% reduction was observed. However, the combination of both viruses at MOI 5 turned out to be a less efficient setup, since only a minor reduction of 8% was perceivable in the fluorescence intensity. There are several possible reasons behind the relatively low efficiency of silencing observed with the virus-based inducible system in comparison to the results of the plasmid experiments. First, as the plasmid-based mimic vectors silence the target eYFP in a dose-dependent manner, the ratio between the target and the mimic is essential. However, the inducible system utilizes the T7 promoter for the transcription of the mimics and the ie1 promoter for eYFP, which changes the relative amounts of the silencer and the target. Second, a prerequisite for the expression of the mimics is the presence of a functional T7 RNAP in the same cell. However, the viruses are added concurrently to the insect culture, thus the RNAP and the target eYFP are being expressed simultaneously. This leads to a certain amount of pre-existing target protein, before the processing of the mimic. Furthermore, the eYFP is known to be a rather stable protein [35], which also contributes to the observed effects. Nevertheless, the flow cytometry data on the protein level indicate the functionality of the inducible system on a viral basis, which was further confirmed on the RNA level with real-time quantitative PCR (RT-qPCR).

### 2.4. Evaluation of the Inducible System on the RNA Level

To reveal, whether the reduction in overall eYFP fluorescence intensity was indeed the effect of specific downregulation, the processing and presence of mature amiRNAs was confirmed (see Figure A2) and the decrease in the eYFP mRNA level was quantified by RT-qPCR. Again, samples were obtained from co-infected (*Ac*-T7amiR-1C_ ie1eYFP or *Ac*-T7amiR-1Cs_ ie1eYFP virus at MOI 1 or 5 in combination with *Ac*-ie1T7RNAP virus at MOI 5 or 10) *Sf*9 cultures. After total RNA extraction and genomic DNA removal, a one-step reverse transcription and RT-qPCR reaction was carried out using specific primers for eYFP. The data were evaluated using the 2^-ΔΔCq^ method [36]. For the calculation of ΔΔCq values and corresponding fold change values presented in Figure 5, ΔCq control values originating from co-infections with the construct-specific scrambled control virus *Ac*-T7amiR-1Cs_ ie1eYFP (ΔΔCq = 0 or fold change of −1) under the same conditions were applied. The results were consistent with those from flow cytometry analysis. The co-infection with 5-fold excess (MOI 5) of the *Ac*-ie1T7RNAP virus compared to *Ac*-T7amiR-1C_ie1eYFP (MOI 1) resulted in a ΔΔCq value of −0.34 corresponding to a 1.3-fold reduction in the eYFP mRNA level, whereas applying the RNAP virus in 10-fold excess (MOI 10) lead to a stronger decrease with a ΔΔ*C*q value of −0.59, indicating a 1.5-fold reduction in eYFP mRNA. However, the co-infection containing the same amounts of both viruses (MOI 5) did not result in a statistically significant suppression in the target mRNA level with a ΔΔ*C*q value of −0.11 and a corresponding 1.1-fold reduction.

## 3. Materials and Methods

### 3.1. Insect Cells and Culture Conditions

*Sf*9 cells (ATCC CRL-1711) were propagated in HyClone SFM4 insect cell medium (GE Healthcare, Little Chalfont, UK) supplemented with 0.1% Pluronic F68 (Sigma-Aldrich, St. Louis, MO, USA). Fifty-milliliter suspension cultures were cultivated in 500 mL flasks at 27 °C with a shaker speed of 100 rpm.

### 3.2. amiRNA Plasmid Constructs

The 600 nt long promoter sequence upstream of the baculoviral immediate-early gene (*ie1*) was PCR amplified using the baculovirus shuttle vector originating from Max Efficiency DH10Bac cells (Invitrogen, Carlsbad, CA, USA) as template. The fragment was then cloned between the ClaI and BamHI restriction sites of the MultiBac acceptor vector pACEBac1 (EMBL, Grenoble, France), thereby replacing the original polyhedrin (polh) promoter and resulting in the pACEBac1ie1 vector. The baculovirus shuttle vector harbored by DH10MultiBacY cells (EMBL, Grenoble, France) was used as template to obtain a PCR fragment of the gene encoding the eYFP (KT878739), which was cloned into the pACEBac1ie1 vector and thus gave rise to the pACEBac1ie1eYFP reporter plasmid.

To facilitate the setup of the inducible system, a special donor vector was created. A 344 nt long fragment containing (in order) the T7 RNAP promoter sequence, 120 nt-s of the 5’ flanking region of the natural miR-1 precursor hairpin [20], a mini multi cloning site (MCS), 120 nt-s of the 3’ flanking region of the pre-miR-1 hairpin, and the TΦ terminator sequence was chemically synthetized by IDT (Leuven, Belgium). After PCR amplification, the product was cloned between the SpeI and PmeI sites of the MultiBac donor vector pIDS (EMBL, Grenoble, France), thus replacing the original cloning cassette and resulting in the pIDST7amiR plasmid. The sequence of the above-described T7amiR fragment embedded in pIDS is presented in Table 1.

As a basis for the amiRNA constructs, the *Ac*MNPV-pri-miR-1 transcript served as a backbone [20]. The original siRNA duplex within the transcript was replaced with a synthetic one containing a siRNA sequence previously proven to be highly effective against its original target eGFP [29]. Furthermore, small changes were applied to the stem sequences, to create overall four altered versions of the *Ac*MNPV-miR-1 hairpin structure: amiR-1A, amiR-1B, amiR-1C, and amiR-1D. In addition to the diversity in the stem-loop structures, there are differences in the length of the flanking regions that were obtained from the natural *Ac*MNPV-pri-miR-1 transcript. Moreover, for each of the amiRNA constructs, a corresponding control was also created by scrambling up the sequence of the given eGFP targeting siRNA duplex incorporated in the amiRNA backbone: amiR-1As, amiR-1Bs, amiR-1Cs, and amiR-1Ds. Sequences of the modified amiRNA hairpin constructs and the scrambled controls are listed in Table 1.

The diverse design of the amiRNA constructs necessitated different cloning procedures for the structures. For amiR-1B, amiR-1C, and amiR-1D (amiR-1B-D) and amiR-1Bs, amiR-1Cs, and amiR-1Ds (amiR-1Bs-Ds), a method described previously was applied. Briefly, each of the stem-loop structures was ordered as two single-stranded, complementary, synthetic oligos from Sigma-Aldrich (St. Louis, MO, USA). The oligos were then pairwise annealed according to the manufacturer´s instructions and subsequently cloned between the BamHI and XbaI sites of the mini MCS in between the 120 nt long flanking regions of the pre-miR-1 hairpin in the donor vector pIDST7amiR, thus resulting in the plasmids pIDST7amiR-1B-D and pIDST7amiR-1Bs-Ds. Furthermore, for the transfection experiments carried out in *Sf*9 cells that served as a preliminary screening of the constructs (see Section 3.3), the backbone of the pACEBac1ie1 vector was used. To this end, the plasmids pIDST7amiR-1B-D and pIDST7amiR-1Bs-Ds served as template for the PCR amplification of the six different fragments, each containing the 120 nt 5´ flank, the stem-loop, and the 120 nt 3´ flank (without the T7 RNAP promoter and terminator sequences). The amplified products were subsequently cloned between the SalI and NotI sites of pACEBac1ie1, giving rise to the plasmids pACEBac1ie1amiR-1B-D and pACEBac1ie1amiR-1Bs-Ds.

The nucleotide sequences encoding the stem-loops amiR-1A and amiR-1As, including the 31 nt long flanking regions of the natural *Ac*MNPV-pri-miR-1 transcript on both sides of the stems, were chemically synthetized as single pieces by IDT (Leuven, Belgium). For the screening experiments, the fragments were cloned, after PCR amplification, between the BamHI and EcoRI sites of the pACEBac1ie1 vector to create the pACEBac1ie1amiR-1A and pACEBac1ie1amiR-1As plasmids, respectively. Furthermore, for the setup of the inducible system, the pIDST7amiR backbone was used and following another PCR amplification, the resulting stem-loop fragments were cloned between the T7 RNAP promoter and terminator sequences (PstI and KpnI sites) of the donor plasmid. This removed the 120 nt 5’ flank, the mini MCS, and the 120 nt 3’ flank necessary for the insertion of the annealed oligos and resulted in the vectors pIDST7amiR-1A and pIDST7amiR-1As. All of the plasmids described here were confirmed with sequencing.

### 3.3. Screening of amiRNA Constructs

The preliminary screening experiments for the estimation of the silencing effectiveness of the different stem-loop constructs comprised of the transfection of insect cells followed by visual estimation by fluorescence microscopy and subsequent flow cytometry analysis. *Sf*9 cells were seeded to 6-well plates with a density of 9 × 10^5^ cells/well and were then pairwise co-transfected with 200 ng of the reporter plasmid pACEBac1ie1-eYFP in combination with 2 µg of one of the eight following plasmids: pACEBac1ie1amiR-1A-D or pACEBac1ie1amiR-1As-Ds. The co-transfections were done with FuGene HD transfection reagent (Promega, Madison, WI, USA) according to the manufacturer’s instructions. Forty-eight hour post-transcription, the eYFP fluorescence intensity was first evaluated using a Leica DM IL LED Inverted Laboratory Microscope and the Leica Application Suite v4.6 software (Leica Microsystems, Wetzlar, Germany). After harvesting the cells, flow cytometry analysis was carried out using a Gallios Flow Cytometer (Beckman Coulter, Vienna, Austria). For the evaluation of the raw data, the Kaluza1.2 software (Beckman Coulter) was applied. All co-transfection experiments were repeated thrice.

### 3.4. Cloning of the Bacteriophage T7 RNA Polymerase

The *T7 gene1* encoding the bacteriophage T7 RNAP (AM946981) was PCR amplified using the lambda DE3 prophage as a template present in *Escherichia coli* BL21(DE3) cells (New England Biolabs, Ipswich, MA, USA). To target the mature RNA polymerase into the nucleus of *Sf*9 cells, where it is needed for the generation of pri-miRNA transcript mimics harboring the amiRNAs, the forward primer used for the PCR amplification contained extra 36 nt encoding the SV40 T antigen nuclear location signal [23,37]. The obtained fragment was cloned within the BamHI and XbaI sites of the pACEBac1ie1 vector, resulting in the pACEBac1ie1T7RNAP construct.

To generate the *Ac*-ie1T7RNAP recombinant *Ac*MNPV, the pACEBac1ie1T7RNAP vector was transformed into Max Efficiency DH10Bac competent cells (Invitrogen). The purified bacmid DNA was then transfected into *Sf*9 cells with FuGene HD transfection reagent (Promega) according to the manufacturer’s instructions. The amplified viral stock of passage three was used to determine the viral titer by plaque assay.

### 3.5. Setup of the Inducible Knockdown System

Acceptor–donor fusion constructs were generated via Cre-LoxP recombination by merging the reporter plasmid pACEBac1ie1eYFP with either pIDST7amiR-1C or pIDST7amiR-1Cs, resulting in the T7amiR-1C_ ie1eYFP and T7amiR-1Cs_ ie1eYFP vectors, respectively. By transforming the fusions into Max Efficiency DH10Bac competent cells (Invitrogen) with FuGene HD transfection reagent (Promega) according to the manufacturer’s instructions, the *Ac*-T7amiR-1C_ ie1eYFP and *Ac*-T7amiR-1Cs_ ie1eYFP recombinant viruses were created. The titers of the amplified viral stocks of passage three were determined by plaque assay.

The silencing of eYFP using the inducible viral system was evaluated on the protein level with flow cytometry. To this end, *Sf*9 cells seeded into T25 flasks at a cell density of 2.5 × 10^6^ cells/flask were co-infected with *Ac*-ie1T7RNAP together with the *Ac*-T7amiR-1C_ ie1eYFP or *Ac*-T7amiR-1Cs_ ie1eYFP virus at various MOI combinations. Forty-eight hours post-infection, the cells were harvested and the eYFP fluorescence intensity was measured with a Gallios Flow Cytometer (Beckman Coulter). For the data analysis, the Kaluza1.2 software (Beckman Coulter) was used. The co-infections were set up three times.

### 3.6. Real-Time Quantitative PCR

Total RNA was extracted from 1 × 10^6^ cells with TRIzol Reagent (Invitrogen) according to the manufacturer’s instructions. The genomic DNA contamination of the RNA samples was removed with the TURBO DNA-free Kit (Invitrogen). Reverse transcription and subsequent RT-qPCR was carried out in a single reaction with the Luna Universal One-Step RT-qPCR Kit (New England Biolabs) according to the manufacturer’s instructions. For the quantification of eYFP mRNA levels, sequence-specific primers (F: 5’-GGCACAAGCTGGAGTACAAC-3’; R: 5’-AGTTCACCTTGATGCCGTTC-3’) that were designed with the GenScript Real-time PCR Primer Design online software were used. As an internal reference gene, the insect 28S rRNA [38] was applied (F: 5’-GCTTACAGAGACGAGGTTA-3’; R: 5’-TCACTTCTGGAATGGGTAG-3’). RT-qPCR was performed in 20 µL reactions consisting of 10 µL of Luna Universal One-Step Reaction Mix (2×), 1 µL of Luna WarmStart RT Enzyme Mix (20×), 0.8 µL of forward primer (10 μM), 0.8 µL of reverse primer (10 μM), 5 ng of DNA-free total RNA template, and nuclease-free water (fill up to 20 µL). The experiments were conducted on a BioRad C1000 Thermal Cycler in combination with a CFX96 Real-Time PCR Detection System (Hercules, CA, USA) using the following program: reverse transcription at 55 °C for 10 min, initial denaturation at 95 °C for 1 min, 40 cycles of denaturation at 95 °C for 10 s, and extension at 60 °C for 30 s (with plate read). Specific amplification was confirmed through melting curve analysis. Each co-infection experiment was repeated three times, and the data were analyzed using the 2^−ΔΔ*C*q^ method [36]. For the evaluation of the statistical significance, the Student´s *t*-test was applied (*p* ˂ 0.05).

## 4. Conclusions

In the framework of this study, a novel inducible knockdown system was established and evaluated in *Sf*9 cells. *Ac*MNPV-miR-1 pri-miRNA transcript mimics [20] harboring an artificial miRNA (targeting eYFP) were designed and assessed with regard to their silencing efficiency. According to plasmid-based screening experiments, the most potent construct was selected for the subsequent evaluation on a viral basis. The inducibility of the viral system lies in the fact that the T7 promoter is inactive in the absence of the prokaryotic T7 RNAP. Thus, separate viruses were created either carrying the polymerase under control of a baculoviral promoter (ie1) or encoding the pri-miRNA transcript mimic accommodating an amiRNA regulated by the T7 promoter. The presence of mature amiRNAs was verified in *Sf*9 cultures co-infected with these viruses. Flow cytometry measurements revealed a reduction of approximately 30–40% in the target eYFP’s overall fluorescence intensity in case the RNAP virus was applied in 5–10-fold excess as compared to the virus carrying the amiRNA. RT-qPCR measurements revealed a decrease in the mRNA level of eYFP, confirming the silencing effect and the functionality of the inducible system. The main advantage of the system is that even genes that are essential for baculovirus propagation can be downregulated. As soon as the two viruses are combined, any desired silencing effect can be triggered. Furthermore, since this system is based on the heterologous T7 expression system, the transcription of functional amiRNA sequences becomes independent of cell-specific PolIII promoters and thus becomes a versatile tool for all animal cells.

## Figures and Tables

**Figure 1 ijms-20-00533-f001:**
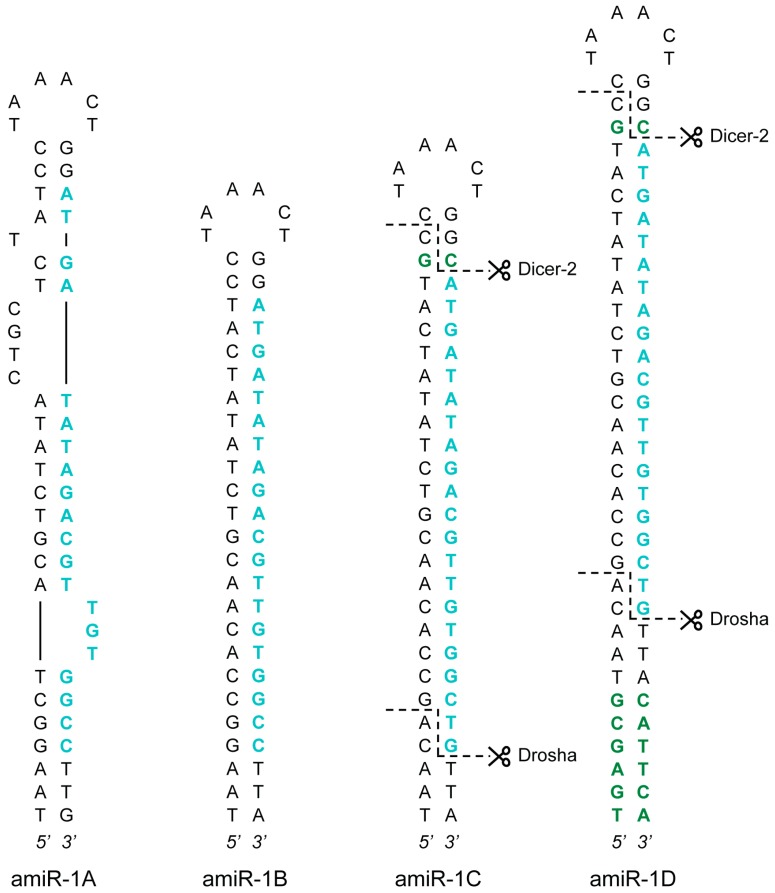
amiRNA precursor hairpin structures. Four structural versions of the *Ac*MNPV-pri-miR-1 transcript were created: amiR-1A-D. The guide strands targeting eYFP are highlighted in blue, whereas the additional nucleotides in the stem regions (in comparison to the *Ac*MNPV-pre-miR-1 hairpin) are showed in green. The anticipated Drosha and Dicer-2 cut sites (as a result of the extra nucleotides) are marked with dashed lines in the amiR-1C and amiR-1D structures.

**Figure 2 ijms-20-00533-f002:**
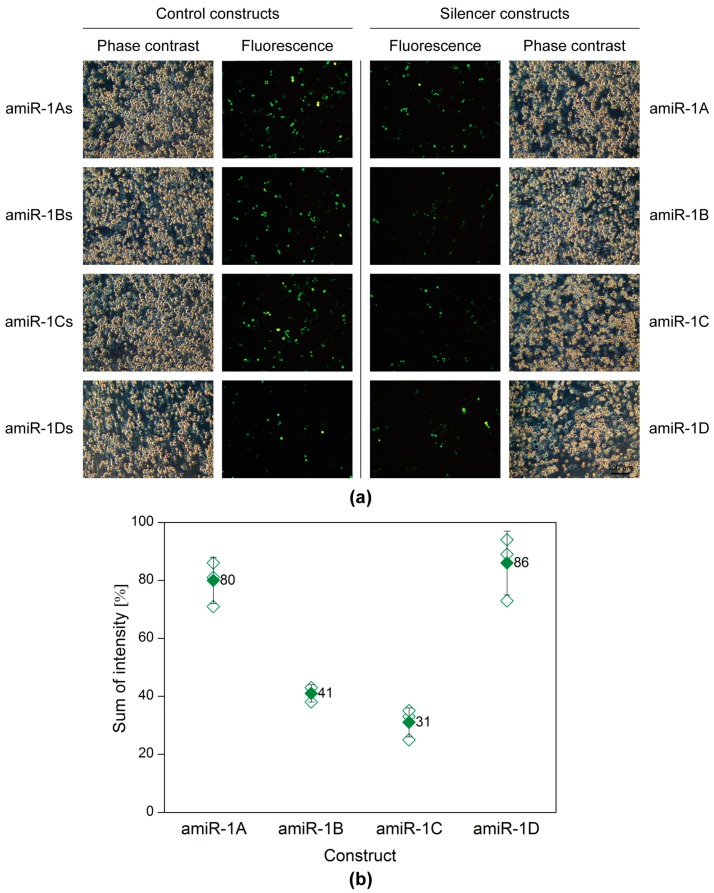
Plasmid-based screening of the amiRNA constructs with fluorescence microscopy (**a**) and flow cytometry (**b**). *Sf*9 cells were co-transfected with the reporter plasmid pACEBac1ie1-eYFP (200 ng) in combination with one of the four artificial hairpin constructs (pACEBac1ie1amiR-1A-D; 2 µg) or the four controls (pACEBac1ie1amiR-1As-Ds; 2 µg). The silencing efficiency was evaluated at 48 h p.t. (**a**) The microscopy images (100× magnification) imply that the highest silencing activity was achieved by amiR-1B and amir-1C, whereas amir-1A and amiR-1D seemed to have no remarkable effect on the overall eYFP fluorescence, when compared to the structure-specific scrambled controls. (**b**) Data obtained by flow cytometry show that amiR-1B and amir-1C reduced the overall eYFP fluorescence by 59% and 69%, respectively. In comparison, amiR-1A and amiR-1D resulted only in fluorescence reductions of 20% and 14%, respectively. The hollow diamonds represent values of the individual measurements, whereas the full diamonds stand for the average sum of intensity value for a given construct.

**Figure 3 ijms-20-00533-f003:**
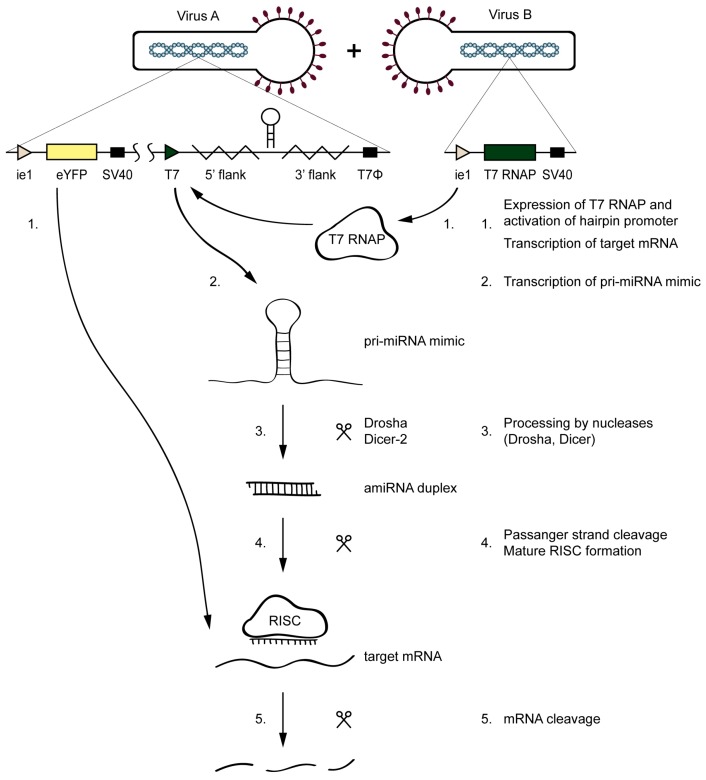
Mechanism of the inducible silencing system targeting eYFP. Upon the co-infection of the *Ac*-T7amiR-1C_ ie1eYFP virus (Virus A) and the *Ac*-ie1T7RNAP virus (Virus B), the T7 RNAP expressed from Virus B (**1**) transcribes the pri-miRNA transcript mimic harboring the eYFP targeting amiR-1C on Virus A (**2**). The transcript enters the host’s miRNA biogenesis pathway, where it is processed by the Drosha and Dicer-2 nucleases (**3**). The resulting amiRNA duplex provides the guide strand (**4**) and thereby activates the RNA-induced silencing complex (RISC) that cleaves the perfectly complementary mRNAs of eYFP (**1**,**5**), thereby impeding its translation and the product formation. No silencing occurs in the control reaction, where the *Ac*-T7amiR-1Cs_ ie1eYFP is used as Virus A.

**Figure 4 ijms-20-00533-f004:**
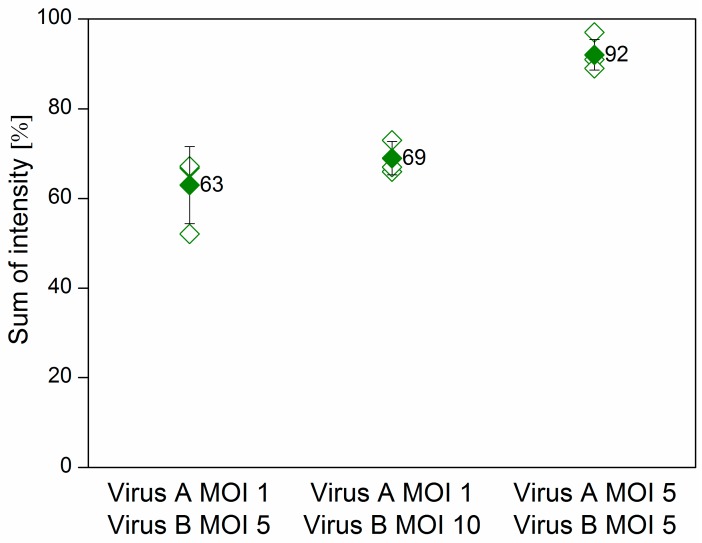
Flow cytometry results of the virus-based inducible system. *Sf*9 cells were co-infected with the fusion virus *Ac*-T7amiR-1C_ ie1eYFP (Virus A) at MOI 1 in combination with the *Ac*-ie1T7RNAP virus (Virus B) in either 5-fold or 10-fold excess (MOI 5 or MOI 10), resulting in a reduction of 37% and 31% in the overall fluorescence intensity, respectively. In case both viruses were added at MOI 5 to the culture, only a slight decrease of 8% was observable. All values are calculated and compared to co-infections with the construct-specific scrambled control virus *Ac*-T7amiR-1Cs_ ie1eYFP (100%) under the same conditions. All co-infections were repeated three times. The hollow diamonds represent values of the individual measurements, whereas the full diamonds stand for the average sum of intensity value for a given construct.

**Figure 5 ijms-20-00533-f005:**
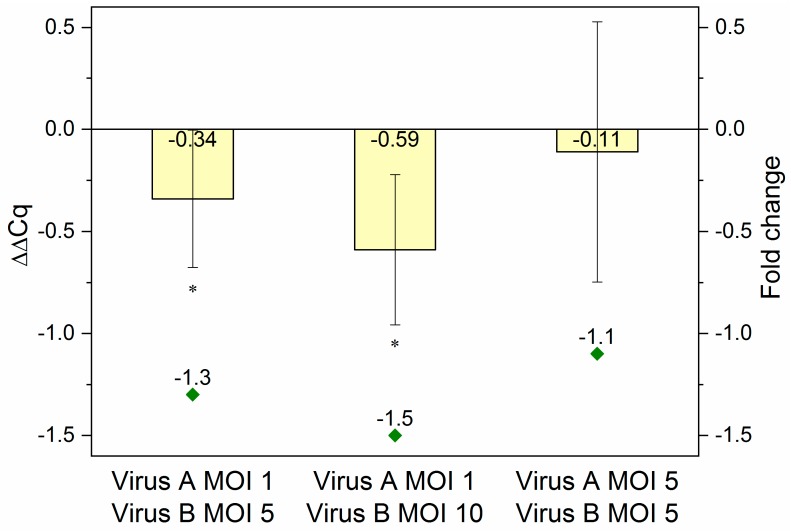
RT-qPCR results. RNA samples originating from *Sf*9 cells co-infected with *Ac*-T7amiR-1C_ ie1eYFP virus (Virus A) at MOI 1 or MOI 5 in combination with *Ac*-ie1T7RNAP virus (Virus B) at MOI 5 or MOI 10 were used for RT-qPCR analysis. In case a lower MOI of 1 was applied from the fusion virus *Ac*-T7amiR-1C_ ie1eYFP together with a 5-fold (MOI 5) or 10-fold (MOI 10) excess of the *Ac*-ie1T7RNAP virus, ΔΔ*C*q values of –0.34 (fold change –1.3) and –0.59 (fold change –1.5) were observed, respectively. The co-infection with MOI 5 of both viruses yielded only a slight reduction in the eYFP mRNA level with a ΔΔ*C*q value of –0.11 (fold change –1.1). All co-infections and RT-qPCR measurements were repeated three times. Statistically significant differences in comparison to the construct-specific scrambled control infections with *Ac*-T7amiR-1Cs_ ie1eYFP virus under the same conditions: * *p* ˂ 0.05.

**Table 1 ijms-20-00533-t001:** Sequences used for the cloning of the artificial miRNA constructs.

Fragment Name	Nucleotide Sequence 5’ to 3’
T7amiR fragment	TAATACGACTCACTATAGGGCTGCAGGTCTATAGATAGCGGTTTTTCGGCAATATACACTTGGCTCAATTTATTATCGCCGTGTGCGATGCGCAAGTTGGCCACCCGGCCGTTATTCAGCTTTACGTTTAATTGTTTGTTCTCGTC*ggatccgaattcctcgagtctaga*AAATTTAATGCATTCGTCCAATAAAGATAAAACAGTATGAGCAAAACGATAAGTAACACGATTCCCCACATGATTTGTTTTAATTTACAATTTCAATTCCAATGAGATTTAGGTTGTGCA*GGTACC*CTAGCATAACCCCTTGGGGCCTCTAAACGGGTCTTGAGGGGTTTTTTG
amiR-1A hairpin construct	TCAGCTTTACGTTTAATTGTTTGTTCTCGTCTAAGGCTACGTCTATACTGCTCTATCCTAAACTGGATGATATAGACGTTGTGGCCTTGAAATTTAATGCATTCGTCCAATAAAGATAAA
amiR-1As hairpin construct	TCAGCTTTACGTTTAATTGTTTGTTCTCGTCTAAACACTCTCAGTAACTGCGACTCCCTAAACTGGGATCTTACTGAGACAGGTGTTTGAAATTTAATGCATTCGTCCAATAAAGATAAA
amiR-1B hairpin construct	*ggatcc*TAAGGCCACAACGTCTATATCATCCTAAACTGGATGATATAGACGTTGTGGCCTTA*tctaga*
amiR-1Bs hairpin construct	*ggatcc*TAAACACTCTCTCGGGTAAAATCCCTAAACTGGGATTTTACCCGAGAGAGTGTTTA*tctaga*
amiR-1C hairpin construct	*ggatcc*TAACAGCCACAACGTCTATATCATGCCTAAACTGGCATGATATAGACGTTGTGGCTGTTA*tctaga*
amiR-1Cs hairpin construct	*ggatcc*TAAACACCTCTCTCAGGTAAAATCGCCTAAACTGGCGATTTTACCTGAGAGAGGTGTTTA*tctaga*
amiR-1D hairpin construct	*ggatcc*TGAGCGTAACAGCCACAACGTCTATATCATGCCTAAACTGGCATGATATAGACGTTGTGGCTGTTACATTCA*tctaga*
amiR-1Ds hairpin construct	*ggatcc*TGAGCGTAAACACCTCTCTCAGGTAAAATCGCCTAAACTGGCGATTTTACCTGAGAGAGGTGTTTACATTCA*tctaga*

The restriction enzyme sites are in italic small caps. The guide strands of the synthetic siRNA duplexes embedded in the amiRNA hairpins are underlined.

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
