# Peer review of "Development of a Dual-Vector System Utilizing MicroRNA Mimics of the Autographa californica miR-1 for an Inducible Knockdown in Insect Cells"

_ijms, 2019, doi:10.3390/ijms20030533_

Reviewer 1 Report

In the manuscript Koczka et al. describe development of new system for downregulation of selected gene in Sf9 cells. My main concern are related to far-fetched, in my opinion, conclusions which are made based on testing of one target and four constructs. I am also not convinced whether obtained knockdown level are sufficient for the purposes of potential use of these constructs.

Specific comments:

-        Manuscript is too long, some parts of results description or methods which contain very detailed information could be placed into supplementary files to make the study description more concise for the reader.

-        Is the suppression by 30-40% expected maximum which can be obtained with this system. Is It enough for a system described as “knockdown”? Is it expected to be enough to improve some production processes?

-        Description of Ago2 activity within RISC is not enough specific. Sometimes it is described as “destroying” complementary sequences, or “digestion”, also this part in Figure 4 is confusing. It should be described as cleavage leading to degradation.

-        Conclusions based on comparison of the activity of four constructs targeting YFP are too general. It cannot be sure that the profile of activity of these precursor would be the same if other siRNA was placed in the stem of these shRNAs

-        Did the authors checked if any isomirs are generated from these constructs? Predicted sites (Figure 1) may not be the only ones or even not a dominant ones. Heterogeneous cleavages of Drosha or Dicer are quite common. This could affect the silencing activity.

-        Additional lines could be included in Figure 3 to make sure with band corresponds to for example 900 nt.

-        More concentrated agarose should be used for resolution of products shown in Figure 6.

-        Presentation of RT-qPCR data in Figure 7 is not clear and confusing. Delta delta Cq is important value for calculations to be made for expression level estimation, but actual changes (FC) should be more clearly presented.

Author Response

Reviewer 1:

Comments and Suggestions for Authors

In the manuscript Koczka et al. describe development of new system for downregulation of selected gene in Sf9 cells. My main concern are related to far-fetched, in my opinion, conclusions which are made based on testing of one target and four constructs. I am also not convinced whether obtained knockdown level are sufficient for the purposes of potential use of these constructs.

Specific comments:

1.      Manuscript is too long, some parts of results description or methods which contain very detailed information could be placed into supplementary files to make the study description more concise for the reader.

The section “2.3. Expression of the bacteriophage T7 RNA polymerase” was combined with parts of the section “3.4. Cloning, detection and in vitro activity assay of the bacteriophage T7 RNA polymerase” and was moved to the Appendix. Furthermore, the first part of the section “2.5. Evaluation of the inducible system on the RNA level” until Figure 6 and the section “3.6. Detection of mature amiRNAs” were also combined and moved to the Appendix.

2.      Is the suppression by 30-40% expected maximum which can be obtained with this system. Is It enough for a system described as “knockdown”? Is it expected to be enough to improve some production processes?

Since our target protein eYFP is a rather stable reporter molecule and yet the system is functional, we expect to observe increased knockdown when targeting other, less stable proteins. In addition, when targeting enzymes, we would expect a 30-40% suppression to lead to more severe reduction in the protein function. Furthermore, several silencer loops targeting different enzymes involved in a certain cellular function can be easily combined and thus an increased overall effect is achievable. We are currently working on the optimization of the system by investigating the effect of certain transcription cassette elements that may increase the knockdown efficiency.

3.      Description of Ago2 activity within RISC is not enough specific. Sometimes it is described as “destroying” complementary sequences, or “digestion”, also this part in Figure 4 is confusing. It should be described as cleavage leading to degradation.

Changes were made accordingly throughout the text and in the figure mentioned by the Reviewer.

4.      Conclusions based on comparison of the activity of four constructs targeting YFP are too general. It cannot be sure that the profile of activity of these precursor would be the same if other siRNA was placed in the stem of these shRNAs.

An extra sentence was added in line 217 mentioning that another construct might be more efficient in combination with another embedded amiRNA sequence.

5.      Did the authors checked if any isomirs are generated from these constructs? Predicted sites (Figure 1) may not be the only ones or even not a dominant ones. Heterogeneous cleavages of Drosha or Dicer are quite common. This could affect the silencing activity.

A common strategy when using short hairpin RNAs is to simply take a precursor miRNA loop backbone and change the internal miRNA sequence while keeping or discarding the internal bulges. Additionally to this, we wished to check whether by applying minimal modifications, a perhaps more effective silencing efficiency could be achieved. It was however not checked whether any isomers were generated from these modified constructs. The generation of the structurally different constructs were based on the currently available information and knowledge on the sequence recognition and cleavage characteristics of Drosha and Dicer. In the figure legend, it is mentioned that the lines represent only the anticipated cleavage sites, not verified ones.

6.      Additional lines could be included in Figure 3 to make sure with band corresponds to for example 900 nt.

Changes were made in the figure accordingly.

7.      More concentrated agarose should be used for resolution of products shown in Figure 6.

The samples were re-run on a 3% agarose gel and the figure was exchanged.

8.      Presentation of RT-qPCR data in Figure 7 is not clear and confusing. Delta delta Cq is important value for calculations to be made for expression level estimation, but actual changes (FC) should be more clearly presented.

Changes were made in the figure accordingly.

Reviewer 2 Report

In the manuscript by Koczka et al, “Development of a dual-vector system utilizing microRNA mimics of the Autographa californica miR-1 for an inducible knockdown in insect cells” the authors first test four artificial microRNA (amiRNA) precursor formats with the same siRNA sequence targeting YFP for knockdown of YFP in Sf9 cells. They then select the most effective format and engineer it into a baculovirus under control of a T7 promoter together with the YFP reporter under control of a baculovirus promoter. In a separate baculovirus they engineer the T7 RNA polymerase under control of a baculovirus promoter. This system allows for the induction of miRNA expression and knockdown of YFP  only when both viruses are used. Using this system the authors show that they can indeed achieve knockdown of YFP with the two viruses and they test different ratios of MOIs.

Overall the manuscript is well written and the results are presented clearly.  I have the following comments to improve the manuscript.

1.       Page 2, line 74.  I am not an expert on the RNAi pathway in insect cells; however, I am quite certain that the TAR RNA-binding protein (TRBP) is not present in insect cells, but rather its function is served by the homologs loquacious (Loqs) and R2D2. Also in Drosophila at least, I think it is Dicer2 not Dicer that is involved in cleaving pre-miRNAs.  The authors should revise this section to ensure that they are using the appropriate names for RNA interference in insect cells, or more specifically Spodoptera frugiperda cells.

2.       Page 4, line 160. The authors say “This however, did not seem to interfere with the processing of the amiRNA.”  Do they have evidence for this.  If yes please refer to the appropriate results, if not this statement should be removed.

3.       Page 6, line 213. I think it is important here for the authors to mention that only one sequence was tested and that the conclusions cannot necessarily be extended to other sequences. Since the actual sequence embedded in the miroRNA scaffold could affect its processing, a different sequence may be more effective in one of the other formats.

4.       Page 6, Figure 2b. It is not clear to me if the data is expressed as a % of each respective control or the average of the controls?  IF it is a percentage of each respective control then I think some data should be presented to show that the controls themselves do not have any effect on the fluorescence.

5.       Page 9, line 289. I think “sequence specific control” should read structurally specific control, or scramble sequence control?

Author Response

Reviewer 2:

Comments and Suggestions for Authors

In the manuscript by Koczka et al, “Development of a dual-vector system utilizing microRNA mimics of the Autographa californica miR-1 for an inducible knockdown in insect cells” the authors first test four artificial microRNA (amiRNA) precursor formats with the same siRNA sequence targeting YFP for knockdown of YFP in Sf9 cells. They then select the most effective format and engineer it into a baculovirus under control of a T7 promoter together with the YFP reporter under control of a baculovirus promoter. In a separate baculovirus they engineer the T7 RNA polymerase under control of a baculovirus promoter. This system allows for the induction of miRNA expression and knockdown of YFP  only when both viruses are used. Using this system the authors show that they can indeed achieve knockdown of YFP with the two viruses and they test different ratios of MOIs.

Overall the manuscript is well written and the results are presented clearly.  I have the following comments to improve the manuscript.

1.      Page 2, line 74.  I am not an expert on the RNAi pathway in insect cells; however, I am quite certain that the TAR RNA-binding protein (TRBP) is not present in insect cells, but rather its function is served by the homologs loquacious (Loqs) and R2D2. Also in Drosophila at least, I think it is Dicer2 not Dicer that is involved in cleaving pre-miRNAs.  The authors should revise this section to ensure that they are using the appropriate names for RNA interference in insect cells, or more specifically Spodoptera frugiperda cells.

The section was revised accordingly; furthermore, “Dicer” was changed to “Dicer-2” throughout the manuscript (including Figures 1 and 4).

2.      Page 4, line 160. The authors say “This however, did not seem to interfere with the processing of the amiRNA.”  Do they have evidence for this.  If yes please refer to the appropriate results, if not this statement should be removed.

The data supporting the observation that the scar sequences did not interfere with the processing of the amiRNA is shown in the Appendix section “Detection of mature amiRNAs” and an additional sentence referring to this data was inserted after the statement.

3.      Page 6, line 213. I think it is important here for the authors to mention that only one sequence was tested and that the conclusions cannot necessarily be extended to other sequences. Since the actual sequence embedded in the miroRNA scaffold could affect its processing, a different sequence may be more effective in one of the other formats.

The paragraph was extended accordingly (line 217).

4.      Page 6, Figure 2b. It is not clear to me if the data is expressed as a % of each respective control or the average of the controls?  IF it is a percentage of each respective control then I think some data should be presented to show that the controls themselves do not have any effect on the fluorescence.

The data presented is the percentage of the averaged control values.

5.      Page 9, line 289. I think “sequence specific control” should read structurally specific control, or scramble sequence control.

The term “sequence specific control” was changed to “scrambled sequence control”.

Round  2

Reviewer 1 Report

Manuscript and presentation of data were largely improved.